# Sinensetin Induces Autophagic Cell Death through p53-Related AMPK/mTOR Signaling in Hepatocellular Carcinoma HepG2 Cells

**DOI:** 10.3390/nu12082462

**Published:** 2020-08-15

**Authors:** Seong Min Kim, Sang Eun Ha, Ho Jeong Lee, Shailima Rampogu, Preethi Vetrivel, Hun Hwan Kim, Venu Venkatarame Gowda Saralamma, Keun Woo Lee, Gon Sup Kim

**Affiliations:** 1Research Institute of Life science and College of Veterinary Medicine, Gyeongsang National University, 501 Jinju-daero, Jinju 52828, Korea; ksm4234@naver.com (S.M.K.); sangdis2@naver.com (S.E.H.); preethivetrivel05@gmail.com (P.V.); shark159753@naver.com (H.H.K.); 2Biological Resources Research Group, Bioenvironmental Science & Toxicology Division, Gyeongnam Branch Institute, Korea Institute of Toxicology (KIT), 17 Jeigok-gil, Jinju 52834, Korea; hojeong.lee@kitox.re.kr; 3Division of Life Sciences, Division of Applied Life Science (BK21 Plus), Plant Molecular Biology and Biotechnology Research Center (PMBBRC), Research Institute of Natural Science (RINS), Gyeongsang National University (GNU), 501 Jinju-daero, Jinju 52828, Korea; shailima.rampogu@gmail.com (S.R.); kwlee@gnu.ac.kr (K.W.L.); 4College of Pharmacy, Yonsei University, Incheon 21983, Korea; gowdavenu27@gmail.com

**Keywords:** sinensetin, autophagy, p53, AMPK, hepatocellular carcinoma

## Abstract

Sinensetin (SIN) has been reported to exhibit anti-inflammatory and anti-cancer activity. However, the cellular and molecular mechanism by which SIN promotes hepatocellular carcinoma (HCC) cell death remains unclear. In the present study, we investigated the induction of cell death by SIN and its underlying mechanism in HepG2 cells, an HCC cell line. We found that SIN significantly induced cell death in HepG2 cells, whereas the proliferation rate of Thle2, human liver epithelial cells, was unaffected by SIN. SIN-treated HepG2 cells were not affected by apoptotic cell death; instead, autophagic cell death was induced through the p53-mediated AMPK/mTOR signaling pathway. Inhibition of p53 degradation led to both autophagy and apoptosis in HepG2 cells. p53 translocation led to SIN-induced autophagy, whereas p53 translocation inhibited SIN-induced apoptosis. However, SIN showed apoptosis in the p53-mutant Hep3B cell line. Molecular docking simulation of the p53 core domain showed effective binding with SIN, which was found significant compared with the known p53 activator, RITA. Collectively, these data suggest that SIN may be a potential anti-cancer agent targeting autophagic cell death in human liver cancer.

## 1. Introduction

Hepatocellular carcinoma (HCC) appears to be highly prevalent and constitutes a significant segment of global cancer mortality, with low levels of survival. According to the 2018 GLOBOCAN estimate, 841,080 new cases of hepatic cancer are predicted to be diagnosed, accounting for 781,631 per year [1]. HCC constitutes more than 90% of liver cancers [2]. Despite advancements in the current diagnosis and major therapies such as chemotherapy and radiotherapy, they also have their own limitations including drug resistance in HCC against anti-cancer drugs, and notably the efficiency and side effects of these therapies are debatable [3]. Hence, there is an urgent need to find an effective therapeutic strategy and agents to treat the uncontrolled cell growth due to deregulation of the natural cell death mechanisms which eliminate mutated cells to develop as cancer cells and cancer progression without causing much destruction to normal cells.

It is well known that progressing proliferative signals and resisting cell death are decisive hallmarks in cancer cells [4]. In addition, it also has the ability to regulate the development and progression of cancer cells by down regulating the growth-stimulating signals when tumor suppressor genes are activated. There is substantial evidence that extracts and compounds derived naturally can suppress tumor development and decrease cancer progression in humans [5]. Initiation of programmed cell death in conventional ways exerts its anti-tumor effect on natural products. Apoptosis is recognized as type I programmed cell death and is usually characterized by distinct morphological characteristics and biochemical mechanisms based on energy. Among the morphological and biological changes that occur in apoptotic cell death, chromatin condensation, nuclear fragmentation, cell shrinkage, and DNA fragmentation are the major features [6]. Apoptosis can be initiated by either the extrinsic death receptor or intrinsic mitochondria-dependent pathways. Unlike necrosis, apoptosis produces cell fragments called apoptotic bodies that phagocytic cells are able to engulf and quickly remove before the contents of the cell can spill out onto the surrounding cells and cause damage [7]. In addition, several natural products have also been reported to contribute to the killing of cancer cells by inducing programmed cell death type II, also called autophagy. Autophagy is the catabolic mechanism of intracellular component degradation to sustain homeostasis and metabolism when cells experience stress, such as nutrient limitations, misfolded protein accumulation, and heat and oxidative stress [8]. The induction of autophagy generally promotes cell survival in cancer cells; however, recent studies have also identified autophagy as a mechanism of cell death. Thus, increasing autophagic cell death is an effective approach to eradicate cancer cells [9]. Studies have reported that AMP-activated protein kinase (AMPK) promoted autophagy by negative regulation of the mammalian target of rapamycin (mTOR) pathway [10]. Attenuation of p53 was recently found to trigger autophagy by phosphorylating AMPK at Thr172 and further inactivation of mTOR [11,12]. Apoptosis and autophagy may interact with each other and be regulated by some common proteins, such as Bcl-2, and p53 [13]. For example, it is well recognized that p53, a tumor suppressor, regulates apoptosis to maintain genomic stability and prevent tumor formation. Reports also indicated that p53 may be involved in the autophagic pathway [14]. Recent studies also revealed that the induction of apoptotic and autophagy cell death depends on the status of p53 and specific cell type [15].

Studies have demonstrated that natural dietary products could represent novel therapeutic options for HCC [5]. Thus, various anti-hepatic cancer strategies have focused on developing natural therapeutic targets over chemotherapy agents [16,17]. Phytochemicals, which are chemical compounds derived from natural herbs, are highly effective for many conditions, including treatment of HCC, with few side effects [18]. In particular, flavonoids have been shown to prevent a variety of human cancers through anti-proliferative and anti-metastasis effects and by promoting programmed cell death [19,20]. Sinensetin (SIN) is one such polymethoxyflavone (PMF) containing five methoxy groups mainly present in citrus fruits. SIN has shown to exhibit numerous biological activities which include anti-inflammatory and anti-cancer activity, as well as enhancement of adipogenesis [16,21,22]. In our previous study, we reported that extracts of edible plants *Citrus platymamma* and *Citrus aurantium* L. contained abundant amounts of SIN with anti-cancer and anti-inflammatory effects [23]. A recent study showed that SIN caused apoptosis and autophagy in human T-cell leukemia [24]. However, the anti-cancer effects and the molecular mechanisms of the SIN compound on HCC cells are not fully understood.

In this research, we investigated the capacity of SIN to attenuate the viability of liver cancer cells and their underlying molecular mechanism. To the best of our knowledge, this research is the first report to elucidate SIN treatment on molecular mechanisms that facilitated autophagic cell death through the p53-mediated AMPK/mTOR pathway in a wild-type p53 HepG2 cell line.

## 2. Materials and Methods

### 2.1. Chemicals and Reagents

SIN was purchased from MedChem Express (MedChem Express, Monmouth Jct., NJ, USA). MG132 (carbobenzoxy-Leu-Leu-leucinal), 3-methyladenine (3-MA), Z-VAD-FMK, and pifithrin-α (PFTα) were purchased from Sigma-Aldrich Co. (St. Louis, MO, USA). Antibodies of caspase-3, poly ADP ribose polymerase (PARP), Bcl-xL, Bak, p53, p-AMPK (Thr172), AMPK, p27, p-mTOR (Ser2448), mTOR LC3B, beclin-1, p62, and lamin B1 were obtained from Cell Signaling Technology (Danvers, MA, USA).

### 2.2. Cell Culture

Normal human liver epithelial cells (Thle2) and HepG2 human hepatocarcinoma cells were obtained from the American Type Cell Collection (Manassas, VA, USA) and the Korean Cell Line Bank (Seoul, Korea), respectively. HepG2 cells were cultured in DMEM (Gibco, Grand Island, NY, USA), and Thle2 cells were cultured in BEGMTM Bulletkit (Lonza, MD, USA) media containing 10% fetal bovine serum (Gibco), 100 U/mL penicillin (Gibco), and 100 μg/mL streptomycin (Gibco) at 37 °C and 5% CO_2_.

### 2.3. Cell Viability and Morphology

Cell viability testing was conducted using the 3-(4,5-dimethylthiazol-2-yl)-2,5-diphenyltetrazolium bromide (MTT; Duchefa Biochemie, Haarlem, Netherlands) method. HepG2 and Thle2 cells were seeded in 48-well plates at a density of 5 × 10^4^ cells per well. The cells were treated with various concentrations of SIN for 24 h or 48 h, followed by the addition of 50 μL 5 mg/mL MTT solution to each well for 3 h at 37 °C. The control group was treated with the same amount of dimethylsulfoxide (DMSO). Disposed medium and the formazan crystals in the cells were dissolved by 500 μL DMSO. Absorbance was measured at 570 nm with a PowerWave HT microplate spectrophotometer (BioTek, Winooski, VT, USA). Cell viability was expressed as a percentage of the control. A morphological analysis of the SIN-treated cells was conducted by phase-contrast microscopy (Olympus, Tokyo, Japan). Nuclear staining with DAPI was executed after 48 h of incubation with SIN at the indicated concentrations. Cells were washed with phosphate-buffered saline (PBS). Fixed cells with 37% formaldehyde and 95% ethanol (1:4) were set for 10 min and washed with PBS and stained with 2.5 μg/mL of DAPI solution for 10 min at room temperature (RT). Then with a florescence microscope, the resulting cells were examined.

### 2.4. Cell Cycle Progression

HepG2 cells were seeded in 6-well plate at a density of 5 × 10^5^ cells, followed by treatment with SIN at different concentrations for 48 h. After the incubation, the cells were collected and fixed with 70% ethanol for 1 h at −20 °C, washed in ice-cold phosphate-buffered saline (PBS) once, then re-suspended in 400 μL of PBS containing 50 μg/mL propidium iodide (PI) and 50 μg/mL RNase A and incubated for 15 min at room temperature in the dark. The cells were immediately analyzed by flow cytometry with a FACS Calibur (BD Biosciences, San Jose, CA, USA) and FlowJo software (FlowJo, Ashland, OR, USA).

### 2.5. Annexin V-Propidium Iodide Apoptosis Detection

Using an APC Annexin-V apoptosis detection kit 1 (BD Pharmingen, San Diego, CA, USA) enabled detection of the apoptotic cells. Shortly after the treatment with various concentrations of SIN (0, 50, and 10 μM) for 48 h, collected cells were washed with PBS and stained with APC-Annexin V and PI for 15 min at room temperature in the dark, in advance of adding binding buffer. Flow cytometry analyses were performed via FACS Calibur (BD Biosciences, San Jose, CA, USA). In each sample, 10,000 cells were sorted.

### 2.6. DNA Fragmentation Assay

The DNA fragmentation was determined using the Apoptotic DNA Ladder kit (Thermo Fisher Scientific; Waltham, MA, USA). HepG2 cells were incubated with different concentrations of SIN for 48 h. Cellular DNA was isolated according to the manufacturer’s instructions, followed by electrophoresis on 1.2% agarose gel (Sigma-Aldrich) containing StaySafe^TM^ Nucleic Acid Gel Stain (Real Biotech Co., Banqiao, Taiwan). DNA was visualized using a UV light and documented by photography.

### 2.7. Detection of Acidic Vesicular Organelles (AVO) with Acridine Orange (AO)

HepG2 cells were seeded at 5 × 10^5^ cells per well in 6-well culture plates and treated with SIN for 48 h. Acidic vacuoles in the HepG2 cells were stained with 1 μg/mL acridine orange (AO; Thermo Fisher Scientific, Waltham, MA, USA) for 15 min and analyzed by flow cytometry with a FACS Calibur (BD Biosciences, San Jose, CA, USA).

### 2.8. Immunoblotting Analysis

The total cell lysates were extracted using RIPA buffer (iNtRON; 50 mM Tris-HCl, pH 7.5, 150 mM sodium chloride, 0.5% sodium deoxycholate, 1% Triton X-100, 0.1% SDS, and 2 mM EDTA) containing a protease inhibitor cocktail and a phosphatase inhibitor (Thermo Fisher Scientific). Protein concentrations were measured using a BCA protein assay kit (Thermo Fisher Scientific). An equal quantity of protein (10 μg) from each sample was electrophoresed on 6–15% SDS-polyacrylamide gels and transferred to polyvinylidene difluoride (PVDF) membranes (Millipore, Bedford, MA, USA). The membranes were blocked using 5% skim milk or bovine serum albumin in TBST buffer (Tris-buffered saline/Tween-20) for 1 h and incubated with primary antibodies at 4 °C for 16 h, followed by HRP-conjugated secondary antibodies (1:500–1:5000; Bethyl, TX, USA) and incubation for 3 h at room temperature. The proteins were detected with ECL reagent (Bio-Rad, Hercules, CA, USA) and analyzed using the Image Lab 4.1 (Biorad) program. The densitometry readings of the bands were normalized according to β-actin expression using Image J software (U.S. National Institutes of Health, Bethesda, MD, USA).

### 2.9. Transmission Electron Microscopy (TEM)

HepG2 cells treated with SIN were fixed for 12 h in 2.5% glutaraldehyde in 0.1 M phosphate buffer (pH 7.4) and washed three times in 0.1 M phosphate buffer. Subsequently, the cells were post-fixed with 1% OsO4 dissolved in 0.1 M phosphate buffer for 2 h at 4 °C. After washing, the cells were dehydrated in graded ethanol (50%–100%) for 15 min each, percolated with propylene oxide, embedded in an epoxy formulation using a Poly/Bed 812 kit (Polysciences, Eppel-heim, Germany), and baked at 60 °C for 48 h. The embedded cells were sectioned in 0.5 μm-thick sections using an ultrasonic disc cutter (Gatan, Model 601, Munich, Germany), stained with 6% uranyl acetate for 15 min, rinsed with distilled water, and then stained with lead citrate (Fisher Scientific, Fair Lawn, NJ, USA). After washing with distilled water, the samples were observed by TEM (Tecnai 12, FEI) at an acceleration voltage of 120 kV.

### 2.10. Preparation of the Protein and Ligands

The protein selected for the present investigation was human p53 core domain bearing the protein data bank (PDB) code 2OCJ [25]. The protein was prepared by removing water molecules and was subsequently minimized. Since this structure lacks the ligand-binding site, the coordinates for the ligand were obtained from 3ZME.The active site residues were defined for all the atoms that fell around the co-crystallized ligand at 10 Å radius. The ligands for the current study were SIN (the ligand of interest) and RITA, a known p53 activator (herein after called the reference compound).

The 2D structures of the ligands were retrieved from the PubChem database (https://pubchem.ncbi.nlm.nih.gov/), and their 3D structures were obtained upon escalating them to Discovery Studio v18. (DS). The ligands were subsequently prepared by minimizing them. Each ligand was allowed to generate 30 conformations upon subjecting them to molecular docking.

### 2.11. Molecular Docking Studies

Molecular docking is one of the superlative approaches that foretells the binding affinities between the protein and the small molecules and further determines its predictive binding mode [26]. For the current investigation, the CDOCKER module available with the DS was employed, and the results were evaluated based upon the -CDOCKER interaction energy. The protein and the ligands were upgraded for docking and the obtained poses were subsequently clustered. From the largest cluster, the poses with highest dock score rendered by key residue interactions were chosen for further analysis.

### 2.12. Statistical Analysis

All experimental results are expressed as the mean ± standard error of the mean (SEM) of triplicate samples. Significant differences between the groups were calculated by one-way factorial analysis of variance (ANOVA) followed by the Bonferroni multiple comparisons test. *P*-values < 0.05 were considered statistically significant.

## 3. Results

### 3.1. SIN Decreases Cell Viability in HepG2 Cells

HepG2 human hepatocellular carcinoma cells and Thle2 human liver epithelial cells were untreated or treated with different concentrations of SIN (Figure 1A; 25, 50, 75, and 100 μM) for 24 h or 48 h, and cytotoxicity was measured by the MTT assay. SIN reduced the viability of HepG2 and Thle2 cells in a concentration- and time-dependent manner. As shown in Figure 1B, 25 μM SIN significantly inhibited the cell proliferation of HepG2 cells at 48 h. Moreover, the cell viability was much lower than in SIN-treated Thle2 cells (Figure 1B). These results suggest that SIN showed cancer cell-specific cytotoxicity and inhibited the proliferation of HepG2 cells.

### 3.2. SIN Has No Apoptotic Effects and Cell Cycle Arrest in HepG2 Cells

Since SIN inhibited cell growth, we examined whether SIN caused apoptosis and cell cycle arrest. To investigate whether SIN induced apoptotic cell death in HepG2 cells, the cells were treated with 0, 50, and 100 μM SIN for 48 h and immunoblots, DNA fragmentation, and cell cycle analysis were performed. The immunoblotting analysis identified that SIN-treatment did not change the protein expression of cleavage of PARP and caspase-3, which are apoptotic protein markers, or Bcl-xL, an anti-apoptotic protein marker (Figure 2A). The data indicated that SIN-treated HepG2 cells did not induce apoptotic cell death. A similar result was obtained from the DNA fragmentation assay. As shown in Figure 2B, DNA fragmentation, one of the hallmarks of apoptosis induction, was not observed in SIN-treated HepG2 cells. An increase in caspase protease activity is essential to apoptotic cell death. HepG2 cells were treated with a pan-caspase inhibitor, z-VAD-fmk, and various concentrations of SIN. As expected, the cell proliferation rate was not restored by treatment with 2 mM z-VAD-fmk and SIN (Figure 2C). The cumulative data indicated that SIN treatment did not induce apoptosis in HepG2 cells. Cell cycle analysis showed no significant changes in HepG2 cell cycles after treatment with SIN, indicating that the inhibition cell proliferation induced by SIN was not due to cell cycle arrest (Figure 2D). The results further indicate that SIN-treated HepG2 cells did not undergo apoptotic cell death and cell cycle arrest.

### 3.3. SIN Triggers Autophagic Cell Death in HepG2 Cells

Natural plant-derived compounds can induce autophagic cell death in human cancer cell lines [27]. We examined whether autophagic cell death was related to the SIN-induced inhibition of cell proliferation. HepG2 cells were treated with indicated concentrations of SIN for 48 h, and the major characteristics of autophagy induction were examined microscopically and by immunoblotting and flow cytometry analysis. A double-membrane vesicle, called an autophagosome, in the cytoplasm characterizes autophagy. To investigate whether SIN induced autophagy in HepG2 cells, morphologic changes were examined by phase-contrast microscopy. Autophagic vacuolization and autophagosome vacuoles were increased in SIN-treated HepG2 cells (Figure 3A). To confirm the morphologic changes observed by phase-contrast microscopy, transmission electron microscopy (TEM) was utilized. An accumulation of cytosolic autophagic vacuoles was observed in SIN-treated HepG2 cells (Figure 3A). The amount of beclin-1 and LC3B-II protein was significantly increased, and the p62 protein related to autophagic flux was dose-dependently decreased by SIN treatment (Figure 3B). The AO staining analyzed by flow cytometry indicated that the formation of AVOs was dose-dependently increased in SIN-treated HepG2 cells (Figure 3C). To estimate the effect of inhibiting autophagy on cell viability, HepG2 cells were treated with SIN in the presence or absence of 3-MA, a selective autophagy inhibitor. The results indicated that co-treatment with SIN and 3-MA significantly improved the cell viability compared to SIN treatment alone (Figure 3D). Our results suggest that the SIN-induced inhibition of cell proliferation in HepG2 cells was due to autophagic cell death.

### 3.4. SIN Treatment Induces Autophagy through the p53-Mediated AMPK/mTOR Pathway in HepG2 Cells

To determine whether p53 was associated with SIN-induced autophagy, p53 protein levels were analyzed by immunoblotting. SIN treatment significantly decreased p53 protein levels compared to untreated cells. The protein expression of p-AMPK and p27 were markedly increased, whereas p-mTOR was decreased in SIN-treated HepG2 cells (Figure 4A). To further investigate whether the SIN-induced autophagy was controlled by the loss of p53, HepG2 cells were treated with SIN in the presence or absence of a protease inhibitor, MG-132, to prevent p53 degradation. Increased LC3B-II, cleaved-PARP, cleaved-caspase-3, and p62 protein expression were observed in the cells co-treated with SIN and MG-132 compared to cells treated with SIN alone (Figure 4B). These results indicate that blockage of SIN-induced autophagy flux was induced by inhibition of p53 degradation. In addition, it shows that inhibition of p53 degradation might have induced in apoptosis in SIN and MG-132 treated HepG2 cells.

### 3.5. SIN Increase Translocation of p53 in HepG2 Cells

To identify the translocation of p53 in SIN-induced autophagy cell death, HepG2 cells were treated with SIN with or without pifithrin-α (PFT-α), a pharmacological p53 transcription in-activator. As shown in Figure 4C, the protein level of cytosolic p53 was decreased, whereas the nucleus p53 level was dose-dependently increased in SIN-treated HepG2 cells, indicating that SIN may control the translocation of p53 with changing total protein level. In addition, co-treatment with SIN and PFT-α prevented translocation of p53 to the nucleus. To demonstrate the p53 translocation effect on autophagy and apoptosis induction, autophagy and apoptosis marker proteins were examined by immunoblotting. In Figure 4D, co-treatment with SIN and PFT-α upregulated p62, cleaved-caspase-3, and cleaved-PARP proteins compared to SIN treatment alone, whereas it downregulated p-AMPK and LC3B-II expression. When p53 translocation was blocked, apoptosis, not autophagy, was induced in SIN-treated HepG2 cells. The results indicate that SIN-induced autophagy was accompanied by the translocation of activation of p53 from the cytosol to the nucleus.

### 3.6. SIN Induces Apoptosis in p53-Null Hep3B Cells

HepG2 cells are the p53 wild-type cell line, and Hep3B are the p53-null cell line [28]. In the current study, p53 played an important role in autophagy induction in SIN-treated HepG2 cells. To further demonstrate the effects of SIN in a p53-null cell line, Hep3B cells were treated with indicated concentrations of SIN, and the cell viability and immunoblotting analysis was done (Figure 5A). The cumulative data indicated that cell viability was decreased more in HepG2 cells than in Hep3B cells (*p* < 0.01 at 100 μM). Immunoblotting analysis indicated that Bak, a pro-apoptotic protein, and cleaved-PARP were increased, and Bcl-xL, an anti-apoptotic protein, was decreased in the SIN-treated Hep3B cells. SIN did not affect LC3B-II and beclin-1 protein expression in Hep3B cells (Figure 5B). To confirm the effect of SIN on the induction of apoptotic cell death in Hep3B cells, apoptosis was examined using annexin V/PI double staining using flow cytometry. As shown in Figure 5C, the results also indicated that 100 μM SIN treatment of Hep3B cells significantly increased early and late apoptotic cell death. These results demonstrated that SIN moderates expression in Hep3B cells.

### 3.7. Evaluation and Validation by Molecular Docking

Molecular docking studies showed that the two ligands occupied the active site, as noticed with the co-crystalized ligand, as seen in Figure 6A. Additionally, it was observed that several residues aided in accommodating the ligands at the binding pocket. The molecular dock scores put forth that the compound RITA demonstrated a -CDOCKER interaction energy of 35.82 kcal/mol, and the compound SIN generated a -CDOCKER interaction energy of 40.36 kcal/mol. These results show that SIN showed a similar binding affinity as that with the known activator.

The reference compound, RITA, generated hydrogen bonds with the residues Val147, Tyr220, and Asp228. The residues Val147 and Pro223 interacted with the pentane rings of the compound by π–alkyl interactions. The residues Val147 and Pro153 rendered carbon hydrogen bonds. Additionally, the residues Leu145, Trp146, Thr150, Pro151, Pro152, Gly154, Thr155, Pro222, Cys229, and Glu221 generated van der Waal’s interactions, accommodating the compound at the binding pocket, as seen in Figure 6B.

The compound SIN was positioned at the active site by several interactions, as depicted in Figure 6A. The residues Pro151, Pro152, Tyr220, Glu221, Ser227, and Thr230 demonstrated carbon hydrogen bonds holding the compound at the active site. Additionally, the residue Tyr220 formed a π–lone pair interaction, and Pro223 displayed π–alkyl interactions with the three rings of the compound. Furthermore, the residues Leu145, Trp146, Val147, Thr150, Pro153, Gly154, Thr155, Pro222, Glu224, Val225, Gly226, and Asp228 held the ligand at the binding pocket via van der Waal’s interactions (Figure 6C).

## 4. Discussion

In recent years, herbal medicine has received distinctive attention as the source of new drugs for anti-cancer therapy [29]. *Citrus platymamma* has been widely used in traditional herbal medicines in Korea, and its flavonoid extracts exhibit various biological effects, including anti-cancer effects [3,30]. SIN is one of the methoxyflavones commonly found in *Citrus* species for various pharmacological effects such as anti-angiogenesis, anti-diabetic, and anti-inflammatory activities [31,32,33]. Although it is often observed in the treatment of several cancers, the effect in liver cancer has not yet been specifically confirmed, and the potential mechanisms are also unknown. In the present study, we first examined the anti-proliferative effect of SIN on different human liver cancer cell lines and normal cell lines. SIN exhibited low cytotoxicity to normal cells (Thle2), while it induced significant cytotoxicity to human liver cancer cells (HepG2).

Flavonoids usually exert their anti-cancer activity by initiating apoptosis, cell cycle arrest, and autophagy in cancer cells [34,35]. Thus, we estimated the cell cycle arrest and apoptosis induction on SIN-mediated cell death. The cell cycle analysis showed there was no significant change in the cell cycle after SIN treatment in HepG2 cells, demonstrating SIN-induced cell death is independent of cell cycle arrest. Annexin V/PI staining for apoptosis, DAPI staining, and DNA fragmentation analysis demonstrated that there was no apoptosis involved in SIN-induced cell death in HepG2 cells. Mitochondria play diverse roles in apoptosis by modulating the pro-apoptotic (Bax) and anti-apoptotic (Bcl-xL) protein ratios that result in caspase activation. Caspases, a family of proteases specific to aspartic acids, are the principal effectors of apoptosis. Caspases works on elimination of cells by restricting the proteolysis of hundreds of substrate proteins [7]. The current study results demonstrated that there was no change in the apoptotic key proteins such as Bcl-xL, caspase-3, and cleaved-PARP, signifying that there was no involvement of apoptosis in SIN-induced cell death on HepG2 cells. An increase in caspase protease activity is essential to apoptotic cell death [36]. That HepG2 cells pre-treatment with z-VAD-fmk was not able to regain the cell viability, which was inhibited on treatment with SIN, confirmed there was no apoptosis, caspase-dependent cell death.

As a corresponding mechanism of cell death, autophagy is also involved in the anti-cancer activity of a wide variety of flavonoids [34,35]. We assessed whether autophagy was activated in SIN-induced cell death. Autophagy starts by forming double-membrane vesicles called autophagosomes that involve cytoplasmic components, organelles, which are submerged following a process of maturation after fusion with lysosomes and eventually become autolysosomes [37]. Autophagic vacuolization and autophagosome vacuoles were increased in SIN-treated HepG2 cells. LC3 proteins are cleaved at the C-terminus by autophagy-related protein 4 (Atg4) to transform LC3-I, which is then conjugated to phosphatidylethanolamine to form LC3-II [38]. During autophagy, beclin-1 and LC3B-II protein play crucial roles in autophagosome establishment. LC3B-II is related to the development of autophagosome membranes [39]. p62 protein levels can be used to investigate the autophagic flux. Autophagy inhibition is associated with upregulation of p62 expression, and autophagy activation is associated with downregulation of p62 level [40]. Our study revealed that SIN remarkably increased the ratio of LC3-II/I and beclin-1 and decreased p62 levels in HepG2 cells. The AO dye fluoresces red in acidic vesicular organelles (AVOs), like autolysosomes, indicating autophagy induction [41]. The AO staining indicated that the formation of AVOs was dose-dependently increased in SIN-treated HepG2 cells. An inhibitory assay in the presence or absence of 3-MA demonstrated that co-treatment with SIN and 3-MA significantly recovered the cytotoxicity relative with only SIN treatment, indicating that the SIN-induced cell death was related to autophagy. Multiple upstream signaling pathways are integrated in the stimulation of autophagy, like AMPK/mTOR, beclin-1 (atg6), endoplasmic reticulum-stress, and calcium signaling [42]. The activation of AMP-activated protein kinase (AMPK) induces autophagy by negative mTOR regulation, and other factors involved in the autophagic pathway control autophagy through AMPK/mTOR signaling. p53 is an indispensable protein in the cellular response to genotoxic and oxidative stress, which inhibits cell growth and proliferation. Accumulated evidence suggests that p53 plays an important regulatory role in the control of autophagy [11]. Autophagy induced by p53 loss was highly related to AMPK activation and mTOR suppression [43]. Cell growth is positively controlled by mTOR activity that can be inhibited by tuberous sclerosis (TSC) complex protein. In the current study, it is noticeable that SIN induced autophagy via the p53/AMPK/mTOR signaling pathway in HepG2 cells. Inhibition of p53 degradation by using MG-132 would have contributed to the occurrence of apoptosis in HepG2 cells.

Recent studies have demonstrated a novel function of p53 localization in autophagy modulation [44,45]. Cytoplasmic p53 was shown to suppress autophagy and promote apoptosis, while nuclear p53 transcriptionally triggered autophagy [46]. Nuclear localization of p53 induced autophagic cell death through the activation of specific genes, such as *DRAM* and *sestrin2* [47]. A recent study showed that p53 translocation to the nucleus led to excessive autophagy by stimulating the AMPK signaling pathway [48].

In the present study, pre-treated HepG2 cells with PFT-α and SIN co-treatment resulted in upregulation of p62, cleaved PARP, and cleaved caspase-3 proteins and further downregulation of p-AMPK and LC3B-II expression as compared with only SIN treatment. Blocking the p53 translocation by PFT-α induced apoptosis rather than autophagy in HepG2 cells treated with SIN. These results suggest that p53 translocations from the cytosol to the nucleus lead to the indication of autophagy (Figure 7). In order to further confirm the role of p53 in SIN-induced liver cancer cell death, a p53-null type Hep3B cell was treated with SIN, which induced apoptosis, while autophagy markers were poorly expressed. The combined results indicate that SIN-induced inhibition of cell viability was higher in HepG2 than in Hep3B cells, and the expression of p53 also plays an important role in SIN-induced cell death in liver cancer. In addition, the molecular docking analysis also demonstrated the effective binding of SIN to the core domain of p53. The computational results demonstrated that SIN interacts with p53 by various important residues, which confirms that SIN can bind directly to p53, leading to cell death.

## 5. Conclusions

This study demonstrates that SIN exhibited a noticeable programmed cell death effect on human hepatocarcinoma cells, which appears to be highly associated with p53 expression. SIN induced autophagy in HepG2 p53 wild type cells, whereas it induced apoptotic cell death in Hep3B, a p53-null cell line. Although SIN showed a small amount of toxicity to Thle2 cells, it was found to be selectively highly toxic to HCC cells, indicating its cancer-specific nature. Our results may have important relevance to the development of strategies for the treatment and prevention of HCC using SIN.

## Figures and Tables

**Figure 1 nutrients-12-02462-f001:**
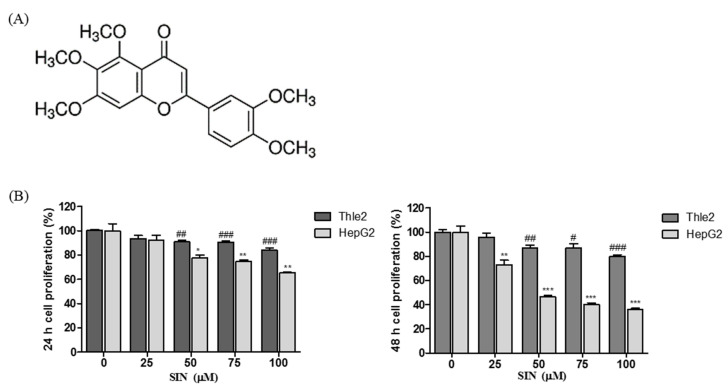
The chemical structure of sinensetin (SIN) and the effect of SIN treatment in HepG2 and Thle2 cells. (**A**) Structure of SIN; (**B**) HepG2 and Thle2 cells were treated with indicated concentrations of SIN (0, 25, 50, 75, and 100 μM) for 24 h or 48 h, and cell viability was determined by the MTT assay. The date are presented as the mean ± SEM of three independent experiments. The results are presented as a percentage compared to vehicle control. # *p* < 0.05 vs. Thle2 cell vehicle control; ## *p* < 0.01 vs. Thle2 cell vehicle control; ### *p* < 0.001 vs. Thle2 cell vehicle control; * *p* < 0.05 vs. HepG2 cell vehicle control; ** *p* < 0.01 vs. HepG2 cell vehicle control; *** *p* < 0.001 vs. HepG2 cell vehicle control.

**Figure 2 nutrients-12-02462-f002:**
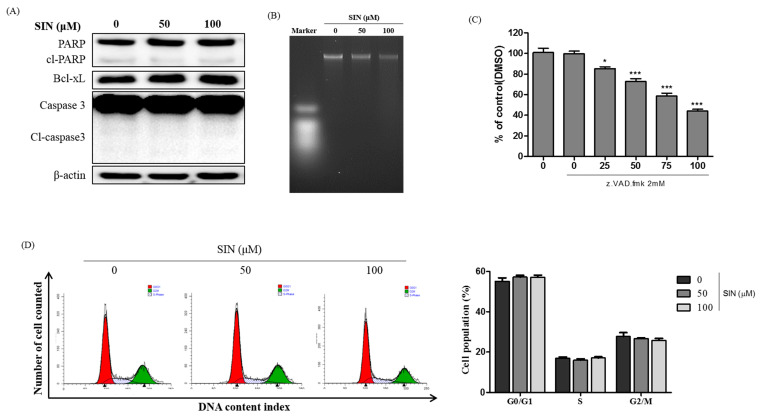
SIN does not induce apoptotic cell death in HepG2 cells. (**A**) HepG2 cells were treated with SIN for 48 h. Immunoblotting analysis of apoptosis markers, cleaved-PARP, cleaved-caspase-3, and Bcl-xL, was performed. β-actin protein was used as a loading control for the analysis. (**B**) The cells were exposed with SIN for 48 h, and then total DNA was isolated and separated on 1.2% agarose gel. (**C**) Cells were exposed to SIN in the presence or absence of 2 mM z.VAD.fmk. Cell viability was examined by the MTT method. (**D**) HepG2 cells were treated with SIN for 48 h, then fixed and stained with propidium iodide (PI). Cell cycle was analyzed with flow cytometry. The mean ± SEM of three independent experiments are expressed as a percentage compared with vehicle control. * *p* < 0.05 vs. vehicle control; *** *p* < 0.001 vs. vehicle control.

**Figure 3 nutrients-12-02462-f003:**
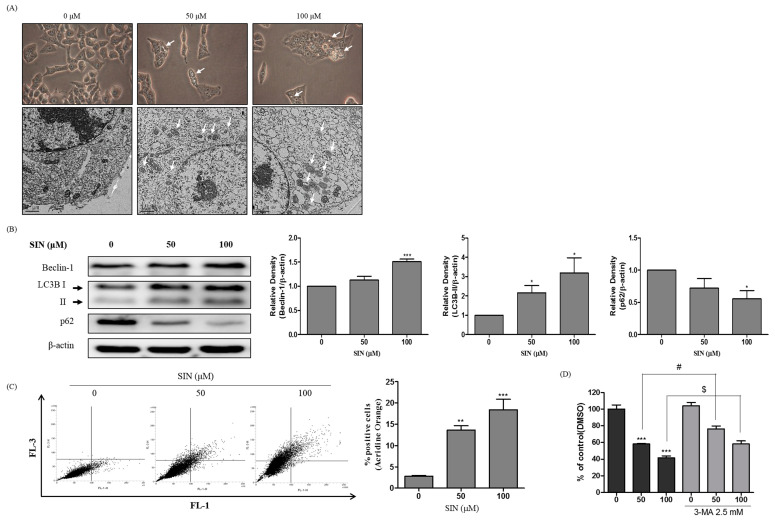
Autophagy is triggered in SIN-treated HepG2 cells. (**A**) The images above shows HepG2 cells treated with SIN for 48 h. The cell morphology was examined by phase-contrast microscopy. The images below shows the autophagic vesicles were visualized by TEM and are indicated by white arrows. (**B**) HepG2 cells were treated with SIN for 48 h. Autophagy markers were investigated by immunoblotting. β-actin was used as a loading control. The quantified expressions by Image J software were plotted in the bar graphs. (**C**) The cells were exposed to SIN for 48 h. Then, the cells were stained with AO to detect AVOs by flow cytometry and the percentage of positively stained cells was calculated. (**D**) The cells were exposed to SIN with or without 2.5 mM 3-MA. Cell viability was analyzed by the MTT assay. The mean ± SEM of three independent experiments is expressed as a percentage compared to vehicle control. * *p* < 0.05 vs. vehicle control; ** *p* < 0.01 vs. vehicle control; *** *p* < 0.001 vs. vehicle control; # *p* < 0.05 vs. compared to the 50 μM SIN-treated group; $ *p* < 0.05 vs. compared to the 100 μM SIN-treated group.

**Figure 4 nutrients-12-02462-f004:**
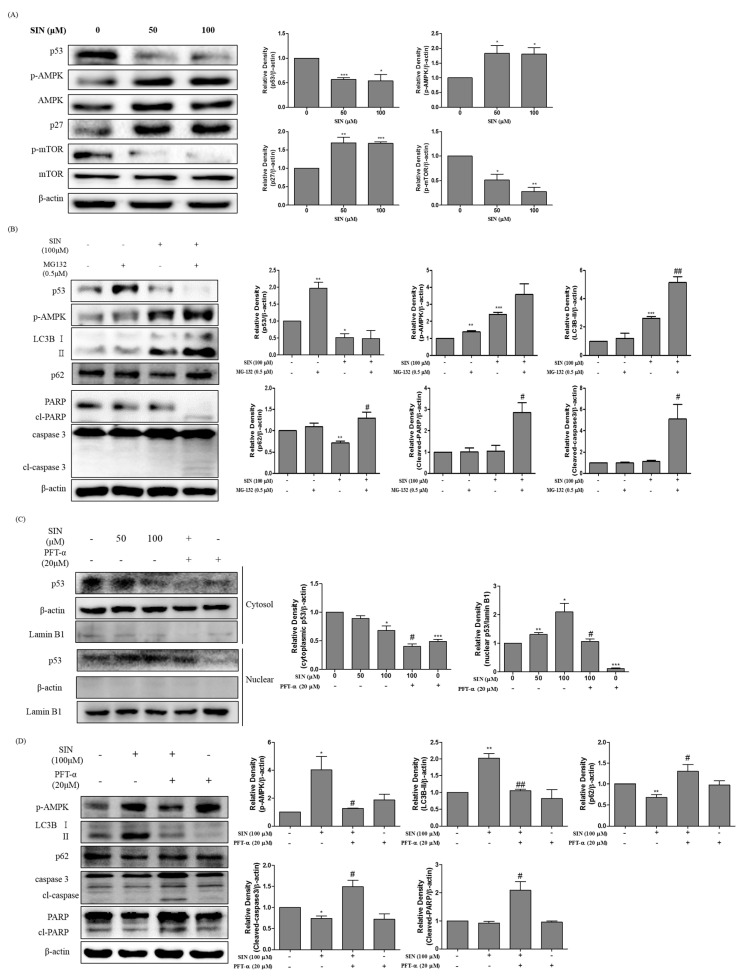
SIN induces the p53-mediated autophagy. (**A**) HepG2 cells were incubated with SIN for 48 h. The protein levels were analyzed by immunoblotting. Bar graphs display the protein/β-actin ratio. (**B**) HepG2 cells were co-incubated with or without SIN and MG132 (0.5 μM) for 48 h. The protein was analyzed by immunoblotting. (**C**) HepG2 cells were incubated with or without SIN and PFT-α (20 μM). The cells were harvested and the cytosolic and nuclear proteins were isolated. p53 translocation was analyzed by immunoblotting. β-actin and lamin B1 proteins were used as cytosolic and nuclear protein loading controls, respectively. (**D**) HepG2 cells were incubated with or without SIN and PFT-α (20 μM) for 48 h. Protein expression was determined by immunoblotting. β-Actin and lamin B1 was used as a loading control and the quantified expressions by Image J software were plotted in the bar graphs. * *p* < 0.05 vs. vehicle control; ** *p* < 0.01 vs. vehicle control; *** *p* < 0.001 vs. vehicle control; # *p* < 0.05 vs. compared to the 100 μM SIN-treated group; ## *p* < 0.01 vs. compared to the 100 μM SIN-treated group.

**Figure 5 nutrients-12-02462-f005:**
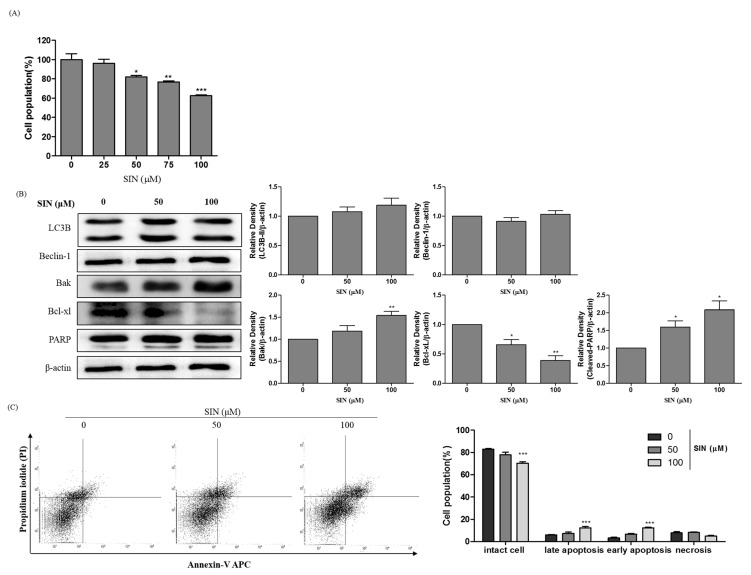
SIN increases apoptosis in p53-null Hep3B cells. (**A**) Hep3B cells were treated with SIN (0, 25, 50, 75, and 100 μM) for 48 h, and the cell viability was determined by the MTT assay. (**B**) Hep3B cells were exposed to SIN for 48 h. Total lysates were analyzed by immunoblotting. β-Actin was used as a loading control, and the quantified expressions by Image J software were plotted in the bar graphs. (**C**) Apoptosis was identified by annexin V-propidium iodide (PI) staining and analyzed by flow cytometry. The data are presented as the mean ± SEM of three independent experiments. The results are presented as a percentage compared to vehicle control. * *p* < 0.05 vs. Hep3B cell vehicle control; ** *p* < 0.01 vs. Hep3B cell vehicle control; *** *p* < 0.001 vs. Hep3B cell vehicle control.

**Figure 6 nutrients-12-02462-f006:**
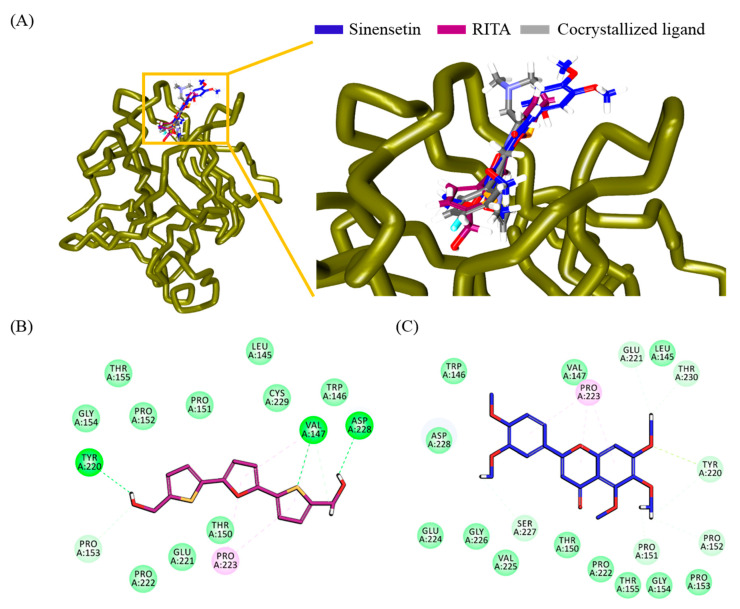
Binding affinity assessment of the ligands at the binding pocket of the target. (**A**) Accommodation of SIN and RITA at the binding pocket of the protein. The left panel depicts the positioning of the ligands at the binding pocket, and the right panel is its zoomed version. (**B**) Overall interactions between the compound RITA and the protein residues. (**C**) Interactions between SIN and protein residues.

**Figure 7 nutrients-12-02462-f007:**
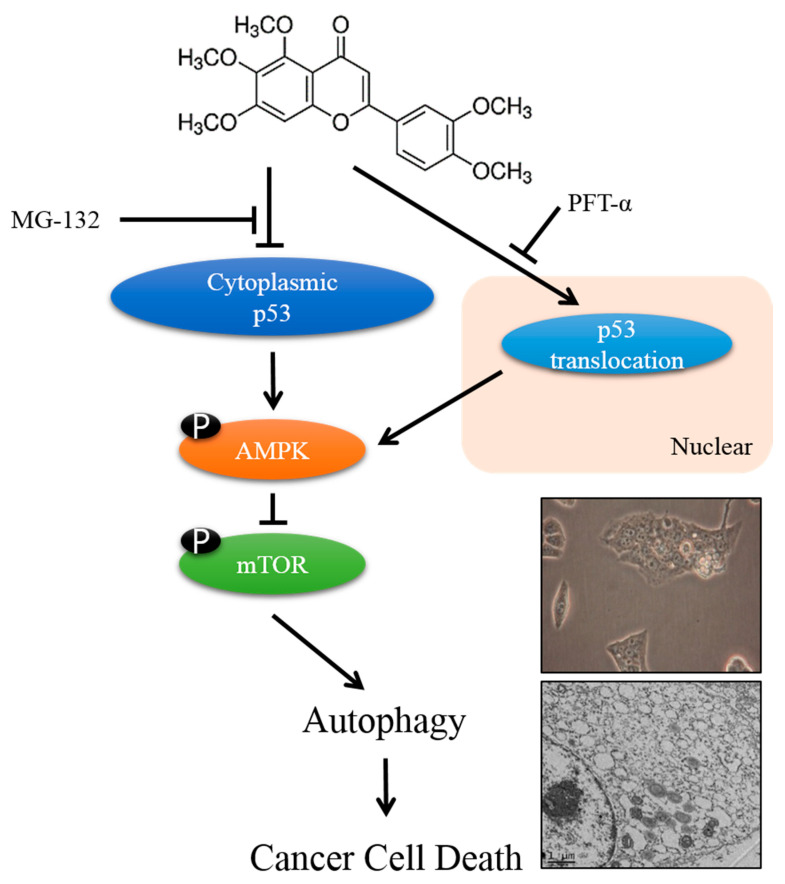
Schematic diagram characterizes the autophagic cell death mechanism of SIN in HepG2 cells.

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
