# Peer review of "Sinensetin Induces Autophagic Cell Death through p53-Related AMPK/mTOR Signaling in Hepatocellular Carcinoma HepG2 Cells"

_nutrients, 2020, doi:10.3390/nu12082462_

Round 1

Reviewer 1 Report

This is a well-designed experiment on the anti-cancer potential of an important natural product which has not been extensively studied. Only minor changes are required:

Line 26, please delete "was" after SIN

Line 31, O do no think it is appropriate to use "a novel mechanism", autophagy is not a novel mechanism, please modify.

Author Response

Reviewer 1.

This is a well-designed experiment on the anti-cancer potential of an important natural product which has not been extensively studied. Only minor changes are required:

Line 26, please delete "was" after SIN

Answer: We removed the word “was” in the mentioned line.

Line 31, O do no think it is appropriate to use "a novel mechanism", autophagy is not a novel mechanism, please modify.

Answer: We removed the word “novel mechanism” followed by your comment.

Reviewer 2 Report

A brief summary

The study by Seong et al. shows that sinensetin (SIN), which is a polymethoxyflavone present in citrus fruits, induces autophagic cell death via p53-AMPK-mTOR pathway in human hepatocellular carcinoma cells.The authors evaluated cell proliferation in Thle2 and HepG2 cells treated with different dose of SIN and concluded that SIN inhibited cell proliferation cancer cell-specifically. In SIN-treated HepG2 cells, cleavage of PARP and caspase 3 was not increased and z-VAD could not block the induction of cell death, suggesting that SIN does not induce apoptosis. While, autophagy markers were drastically changed and 3-MA could rescue this cell death. From further analysis, the authors elucidated that the autophagy was induced through p53-mediated AMPK/mTOR pathway. In Hep3B cells, which lacks p53, SIN seemed to induce apoptosis, not autophagic cell death.

Broad comments

The anti-tumor effect of sinensetin is particularly interesting. For the most part, the experiments are well done and the manuscript is straightforward and well written. However, some of the conclusions are not necessarily supported by the data provided in its current form. In order to further improve the work, it is advisable to consider following points. Major and minor concerns are described below.    

Specific comments

Major points:

  1. In figure 3D, cell viability is still decreased in dose dependent manner even with 3-MA administration. Are there any possibilities that some other types of programmed cell death are induced by SIN?

  1. In Figure 4C, it looks that the total amount of p53 in both cytoplasm and nucleus is decreased by PFT-α treatment rather than PFT-α blocks the translocation of cytoplasmic p53 to nucleus. Are there any other proofs that PFT-α certainly blocks the translocation?

  1. In line 324-325, the authors said that “These results demonstrated that SIN only induced apoptotic cell death in Hep3B cells.” However, the data shown in Figure 5 seems not to be sufficient to support this conclusion. The authors should show that the blockage of apoptotic pathway (e.g. z-VAD-fmk) can rescue this induction of cell death.

Minor points:

  1. In Figure 4C, the legend of SIN concentration in immunoblotting plot is weird. It should be corrected.

  1. In line 317-318, the authors said that “The cumulative data indicated that cell viability was decreased more in HepG2 cells than in Hep3B cells”. In order to conclude this, some statistical analysis should be shown.

  1. The title of Figure 5 contradicts to the statement in line 324-325. It should be adjusted to the main manuscript.

  1. In line 431, unintentional “n” should be deleted.

  1. In line 444, the authors said that “… it shows negligible amount of toxicity towards Thle2 cells, demonstrating its cancer specific activity”. However, this statement seems to be too exaggerated because cell proliferation is still decreased in dose-dependent manner even in Thle2 cells. It might be better to change it to a more moderate expression.

Author Response

Reviewer 2.

Specific comments

Major points:

1.In figure 3D, cell viability is still decreased in dose dependent manner even with 3-MA administration. Are there any possibilities that some other types of programmed cell death are induced by SIN?

Answer: Thank you for your valuable comments. We performed other types of programmed cell death such as necroptosis, paraptosis, and apoptosis in SIN treated HepG2 cells. However, SIN induced only autophagic cell death in HepG2 cells. Therefore, we treated HepG2 cells with 3-MA (autophagy inhibitor) to confirm the induction of autophagy specifically. In figure 3D, the cell viability has not fully recovered in the inhibitor co-treated group. We anticipate that it is because of the efficacy concentration of 3-MA.

2.In Figure 4C, it looks that the total amount of p53 in both cytoplasm and nucleus is decreased by PFT-α treatment rather than PFT-α blocks the translocation of cytoplasmic p53 to nucleus. Are there any other proofs that PFT-α certainly blocks the translocation?

Answer: In the figure 4C, we focused the p53 translocation during PFT-α treatment in HepG2 cells and found that p53 translocation was significantly increased in SIN treatment. PFT-α has been widely used as a specific inhibitor of p53 translocation activity. Then, we compared p53 translocation with PFT-α treatment in presence or absence of SIN. Although total amount of p53 in cytoplasm was decreased in the PFT-α treatment compared with untreated group, we found an increase of p53 translocation in PFT-α and SIN co-treated group compared with the only PFT-α treated group.  

3.In line 324-325, the authors said that “These results demonstrated that SIN only induced apoptotic cell death in Hep3B cells.” However, the data shown in Figure 5 seems not to be sufficient to support this conclusion. The authors should show that the blockage of apoptotic pathway (e.g. z-VAD-fmk) can rescue this induction of cell death.

Answer: We agree with your suggestion. We checked the apoptotic cell death in Hep3B cells using the representative apoptosis related proteins (Bak, Bcl-xl and cleaved PARP). In addition, we performed annexin V and PI staining analysis using flow cytometry to check apoptosis induction. The caspase inhibitor (z-VAD-fmk) experiments is of course more certain, however, we think the existing results are also fairly enough.  

Minor points:

1.In Figure 4C, the legend of SIN concentration in immunoblotting plot is weird. It should be corrected.

Answer: We improved the resolution of figure 4C.

2.In line 317-318, the authors said that “The cumulative data indicated that cell viability was decreased more in HepG2 cells than in Hep3B cells”. In order to conclude this, some statistical analysis should be shown.

Answer: We performed the statistical analysis of cell viability in HepG2 and Hep3B at 100 μM treatment and mentioned in the line 318.

3.The title of Figure 5 contradicts to the statement in line 324-325. It should be adjusted to the main manuscript.

Answer: We changed the title of figure 5.

4.In line 431, unintentional “n” should be deleted.

Answer: We removed the letter“n”.

5.In line 444, the authors said that “… it shows negligible amount of toxicity towards Thle2 cells, demonstrating its cancer specific activity”. However, this statement seems to be too exaggerated because cell proliferation is still decreased in dose-dependent manner even in Thle2 cells. It might be better to change it to a more moderate expression.

Answer: We changed the sentence to “Though SIN shows a small amount of toxicity to Thle2 cells, it is found to be selectively high toxic to HCC cells indicating its cancer specific nature” in the manuscript.

Round 2

Reviewer 2 Report

Basically, the authors have addressed my concerns, other than 2 relatively minor issues.

  1. In Figure 4C, the captions of SIN dose are different between the image of immunoblotting and the bar graph. 
  2. In line 326-327, the authors said that "... SIN only induced apoptotic cell death in Hep3B cells." However, this expression seems to be too exaggerated. It might be better to change to the moderate expression.

Author Response

  1. In Figure 4C, the captions of SIN dose are different between the image of immunoblotting and the bar graph.

*Answer: We changed the Figure 4C image captions of immunoblotting and the bar graph.

  1. In line 326-327, the authors said that "... SIN only induced apoptotic cell death in Hep3B cells." However, this expression seems to be too exaggerated. It might be better to change to the moderate expression.

*Answer: We changed it followed by your comment.
